# The clinical characteristics and risk factors of steroid-induced ocular hypertension following pars plana vitrectomy

Shuanghong Jiang, Hongxia Yang, Ting Chen, Zhenyu Ji, Xixi Yan *

Eye Center, Renmin Hospital of Wuhan University, Wuhan, China

* yanxixi@whu.edu.cn

## Abstract

### Objective

This study aimed to assess the incidence and risk factors for the development of steroid-induced ocular hypertension (SIOH) following 23-gauge (23G) pars plana vitrectomy.

### Methods

The clinical data of patients treated with 23G vitrectomy from January 2019 to March 2022 were reviewed retrospectively. The incidence and characteristics of SIOH post-operatively and treatment provided were recorded. The clinical risk factors for developing SIOH were analyzed using logistic regression analysis.

### Results

Among the 540 eligible patients, 111 (20.56%) cases developed SIOH. The majority (83.78%) of the SIOH cases developed between the third and seventh day postoperatively. Among these cases, 65 (58.56%) patients had an intraoperative pressure (IOP) of 30 mmHg or higher, and 31 (27.9%) had an IOP of 40 mmHg or higher. The IOP of all SIOH patients returned to normal within 1 month following the discontinuation of steroid and IOP-lowering medicine treatment. The independent risk factors for SIOH (IOP ≥ 23 mmHg) were myopia (odds ratio (OR) 5.22) and silicone oil filling (OR 8.20). For severe SIOH (IOP ≥ 30 mmHg) myopia and silicone oil filling were also identified as risk factors with an OR of 3.23 and 12.86, respectively. After adjusting the steroid administration pattern, the incidence of all SIOH and severe SIOH decreased to 17.11% and 9.14%, respectively.

**Data availability statement:** All relevant data are available from the figshare repository at the following DOI: 10.6084/m9.figshare.31442131.

**Funding:** Hubei Provincial International Science and Technology Cooperation Program (2024EHA051).

**Competing interests:** The authors have declared that no competing interests exist.

**Abbreviations:** IOP: intraocular pressure; SIOH: Steroid-induced ocular hypertension; 23G: 23-gauge; OR: odds ratio; CI: confidence interval; SD: standard deviation.

## Conclusions

Myopia and silicone oil filling were identified as potential independent risk factors for the development of SIOH after vitrectomy. A shorter topical steroid therapy was associated with a lower incidence of SIOH particularly in high-risk patients.

## Introduction

Raised intraocular pressure (IOP) is a common postoperative complication of vitrectomy. Topical steroids and antibiotics are often administered after a vitrectomy to reduce inflammation and risk of infection, thereby promoting better healing and visual outcomes. However, the administration of steroids has also been linked with the development of raised IOP [1]. At our local center, we also observed that some patients developed high IOP accompanied by severe eye pain, headache, nausea, and vomiting following the administration of topical steroids. Elevated IOP usually normalizes within a few days after discontinuing the steroid treatment. However, raised IOP can result in optic nerve damage and potentially irreversible vision loss [2]. Factors such as a family history of open-angle glaucoma, myopia, connective tissue disorders, and age have been linked with the development of raised IOP [3].

In recent years at our center, we have replaced the 20-gauge (20G) vitrectomy system with the 23-gauge (23G) system. Compared to the 20G system, the smaller size of the 23G surgical instruments results in less invasive surgery, faster healing, reduced postoperative discomfort, and fewer complications. As a result, the 23G could potentially reduce the need for administering steroid treatment to reduce inflammation post-surgery [4]. Although several studies evaluated the incidence of high IOP after vitrectomy, few studies assessed the risk factors that lead to steroid-induced ocular hypertension (SIOH). Therefore, in this study, we aimed to evaluate the incidence of SIOH following 23G vitrectomy. In addition, we also aimed to identify the risk factors for developing SIOH after 23G vitrectomy and the impact of early steroid discontinuation on IOP in patients at risk of developing SIOH.

## Methods

### Study subjects

The data of adult patients who underwent vitrectomy at our hospital between 01/01/2019 and 31/03/2022 were retrospectively accessed for research purposes from 12/07/2022–11/12/2022 following ethical approval. The exclusion criteria are as follows: preoperative glaucoma or ocular hypertension, ocular trauma, diabetic retinopathy, a family history of glaucoma or history of connective tissue disease, postoperative conditions including exudative anterior uveitis, elevated IOP caused by pupillary block, anterior chamber or vitreous hemorrhage, silicone oil-related factors (e.g., silicone oil entering the anterior chamber, overfilling, or emulsification), gas expansion and incision leakage (defined as IOP < 9 mmHg on the first day postoperative). [5]

## Data collection

The clinical characteristics of the patients including age, gender, medical history, diagnosis, pre-and postoperative IOP, presence of myopia, surgical method, and medication used were collected by reviewing the electronic medical records. In addition, the occurrence of high IOP and treatment after discharge reported at 1 month postoperatively at outpatient clinics were also recorded. Those patients who did not have follow-up reports were called up and asked whether their general practitioner had told them if they had raised IOP.

## Surgical procedure and postoperative management

Most surgical procedures were performed under local retrobulbar anesthesia. Some patients were treated under general anesthesia. A standard scleral incision was made 3.5 mm posterior to the limbus corneoscleralis, and a three-channel closed 23G vitrectomy (BL1433 Stellaris, Bausch&Lomb Incorporated, USA) was performed. During the surgery, phacoemulsification, cryotherapy, intraocular laser photocoagulation, silicone oil filling, or filtered air filling were selectively performed according to the patient's condition. For aphakic eyes, a 6-o'clock peripheral iridectomy was performed intraoperatively.

Patients usually stayed in the hospital for 3–7 days for postoperative observation. After the surgery, patients who received silicone oil or filtered air filling were instructed to rest in the prone position. In addition, patients were also asked to apply tobramycin and dexamethasone eye drops (Tobradex, Alcon Laboratories) three times daily to the operated eye, and tobramycin and dexamethasone ointment (Tobradex, Alcon Laboratories) once every night for 2 weeks after surgery. Patients at risk of developing proliferative vitreoretinopathy were administered systemic prednisone (40 mg daily) for 1–3 days. Postoperative visual acuity, non-contact IOP measurements (Topcon CT-80, Topcon Corporation, Japan), and slit-lamp examinations of the anterior and posterior eye segments were performed. The patients were reviewed 2 weeks after surgery, and the steroid medication was discontinued and replaced with a non-steroidal anti-inflammatory eye drop.

## Management of SIOH

The IOP value was recorded as the average of three valid measurements. SIOH was defined as an IOP of 23 mmHg or higher that developed on the second days post-surgery before the discontinuation of tobramycin dexamethasone and controlled within 1 month following the discontinuation of tobramycin dexamethasone and administration IOP-lowering drugs. Patients with SIOH (23 mmHg to 40 mmHg) were treated by discontinuing the use of tobramycin dexamethasone and the administration of topical IOP-lowering drugs including carteolol hydrochloride, brimonidine tartrate, and brinzolamide, while those with severe SIOH (IOP > 40 mmHg) were treated with topical IOP-lowering drugs and an additional dose of 20% mannitol intravenously.

The risk factors for developing SIOH were identified by reviewing the data of SIOH patients treated with traditional steroids between January 2019 and July 2019 at our center. Based on these findings, all patients treated after December 2019 with a high risk of developing SIOH and a mild intraocular inflammatory response to steroids received a shorter duration of topical tobramycin and dexamethasone administration according to the patient's needs. The latter patients were identified as the adjusted steroid group.

## Statistical analyses

Univariate and multivariate logistic regression analyses were used to identify the risk factors for developing SIOH under the traditional medication model. The difference in the occurrence of SIOH between the traditional treatment group and the adjusted model was compared using the t-test or chi-square test. All statistical tests were performed using the Stata software version 15.0, and a $P$-value below 0.05 was considered statistically significant.

## Ethical considerations

This research study was approved by the Institutional Review Board (WDRY2022-K123). Written informed consent was waived due to the retrospective nature of this study.

## Results

### Baseline characteristics of the participants

A total of 540 patients were included in this study of whom 201 patients were treated using the traditional steroid method and 339 were treated with adjusted steroids. The average age of the patients was 57.50 ± 12.01 years. Males accounted for 50.56% of the participants. The pre-operative assessment showed that 34.81% were myopic. The average preoperative IOP was 15.86 ± 3.32 mmHg. The main indications for vitrectomy for the patients in this study were rhegmatogenous retinal detachment (54.44%), macular membrane and macular hole (18.52%), retinal vein occlusion with vitreous hemorrhage (9.44%), and isolated vitreous hemorrhage (4.44%).

### Incidence of SIOH post-surgery

A total of 111 (20.56%) patients were diagnosed with SIOH post-surgery. Among them, 65 (58.56%) had an IOP of 30 mmHg and higher, and 31 (27.9%) and IOP of 40 mmHg and higher. The average IOP elevation onset was 4.82 ± 1.89 days and ranged between 2 and 10 days. Notably, 83.78% of the cases occurred between the third and seventh day after surgery. After discontinuing the use of steroid eye drops and administering IOP-lowering medications, the IOP of all patients normalized within 1 month.

### Risk factors for SIOH

The univariate analysis identified myopia, preoperative vitreous hemorrhage, cryotherapy, silicone oil, and gas filling as risk factors for developing SIOH (IOP ≥ 23 mmHg) and myopia, cryotherapy, and silicone oil filling as risk factors for developing severe SIOH (IOP ≥ 30 mmHg) (Table 1).

The multivariate analysis identified myopia (odds ratio (OR) 5.22) and silicone oil filling (OR 8.20) as independent risk factors for SIOH. For severe SIOH (IOP ≥ 30 mmHg) myopia and silicone oil filling were also identified as risk factors with an OR of 3.23 and 12.86, respectively (Table 2).

**Table 1. Univariate logistic regression of risk factors for SIOH in vitrectomy patients.**

| Factors | IOP ≥ 23 mmHg | | IOP ≥ 30 mmHg | |
|---|---|---|---|---|
| | OR (95%CI) | *P* value | OR (95%CI) | *P* value |
| Age (every 20 years) | 0.67 (0.40-1.14) | 0.145 | 0.74 (0.40-1.36) | 0.343 |
| Male vs female | 1.05 (0.55-1.98) | 0.873 | 1.20 (0.57-2.51) | 0.626 |
| Myopia vs non-myopia | 5.92 (2.99-11.74) | <0.001 | 3.89 (1.80-8.38) | 0.001 |
| Vitreous hemorrhage preoperatively | 0.42 (0.18-0.96) | 0.040 | 0.54 (0.21-1.40) | 0.211 |
| Phacoemulsification | 0.85 (0.22-3.21) | 0.812 | 1.51 (0.39-5.84) | 0.543 |
| Cryotherapy | 2.13 (1.10-4.10) | 0.024 | 2.61 (1.17-5.79) | 0.018 |
| Photocoagulation | 1.93 (0.86-4.31) | 0.105 | 2.27 (0.82-6.21) | 0.110 |
| Silicone oil filling | 11.96 (3.56-40.13) | <0.001 | 21.02 (2.80-157.51) | 0.003 |
| Air filling | 0.15 (0.04-0.68) | 0.014 | No case | – |
| Systemic use of steroid | 0.75 (0.31-1.77) | 0.514 | 0.37 (0.10-1.31) | 0.126 |

SIOH = steroid-induced ocular hypertension. IOP = intraocular pressure. OR = odds ratio. CI = confidence interval.

**Table 2. Multiple logistic regression of risk factors for SIOH in vitrectomy patients.**

| Factors | IOP ≥ 23 mmHg | | IOP ≥ 30 mmHg | |
|---|---|---|---|---|
| | OR（95%CI） | *P* value | OR（95%CI） | *P* value |
| Age (every 20 years) | 1.14(0.61-2.10) | 0.670 | 1.17(0.60-2.30) | 0.631 |
| Male vs female | 0.87(0.41-1.90) | 0.743 | 1.22(0.52-2.85) | 0.642 |
| Myopia vs non-myopia | 5.22(2.39-11.40) | <0.001 | 3.23(1.36-7.68) | 0.008 |
| Cryotherapy | 1.26(0.53-22.97) | 0.585 | 1.70(0.64-4.52) | 0.282 |
| Silicone oil filling | 8.20(2.22-30.20) | 0.002 | 12.86(1.61-102.9) | 0.016 |

SIOH = steroid-induced ocular hypertension. IOP = intraocular pressure. OR = odds ratio. CI = confidence interval.

## Comparison between the traditional and adjusted steroid treatment

There was no significant difference in age, preoperative IOP, the proportion of myopia, and the number of cases with silicone oil filling between the two treatment groups at baseline. The patients in the traditional steroid group received an average of 9.80±3.31 days of topical steroids and 0.37±0.89 days of systemic steroids. The patients treated with adjusted steroid therapy received on average a shorter treatment (7.09±3.14 days) of topical steroids and a longer treatment of systemic steroids (1.36±1.77 days) (Table 3). After excluding patients who discontinued topical steroids due to SIOH, the myopic patients in the traditional steroid group received significantly more days of topical steroid treatment (10.86±2.59 days versus 6.42±2.96 days (*P*=0.004)). Similarly, the patients with silicone oil tamponade in the traditional treatment group received more days of topical steroid treatment when compared to the adjusted steroid group (10.88±2.55 days and 7.03±3.06 days *P*<0.001).

As shown in Table 4, when compared with the traditional steroid group, the incidence of SIOH was significantly lower in the adjusted steroid group for both IOP of 23 mmHg or higher (26.37% versus 17.11%, *P*=0.010 and IOP of 30 mmHg or higher (16.92% versus 9.14%, *P*=0.007). The adjusted steroid group also had a longer mean average onset for SIOH than the traditional steroid group (5.26±1.90 versus 4.43±1.80 days *P*=0.02).

## Discussion

SIOH is a common side effect following vitrectomy. Although the IOP of most patients with SIOH can be restored to normal levels by stopping steroid therapy, studies have shown that even a short-term state of high IOP may cause irreversible

**Table 3. Characteristics of the study population before and after control medication duration.**

| Factors | Traditional steroid administration | Adjusted steroid administration | *P* value |
|---|---|---|---|
| Age, mean±SD, years | 57.66±10.93 | 57.41±12.63 | 0.814 |
| Preoperative IOP, mean±SD, mmHg | 16.07±3.28 | 15.74±3.34 | 0.255 |
| Gender, n(%) | | | |
| Male | 114(56.72) | 159(46.90) | 0.027 |
| Female | 87(43.28) | 180(53.10) | |
| Myopic eye | 70(34.83) | 118(34.81) | 0.997 |
| Days of topical steroids using, mean±SD | 9.80±3.31 | 7.09±3.14 | <0.001 |
| Days of systemic steroids using, mean±SD | 0.37±0.89 | 1.36±1.77 | <0.001 |
| Tamponade, n(%) | | | |
| Air | 32(15.92) | 91(26.84) | 0.003 |
| Silicone oil | 135(67.16) | 212(62.54) | 0.278 |

SD = standard deviation. IOP = intraocular pressure.

**Table 4. SIOH related factor before and after changing the topical steroid administration.**

| Factors | All | Traditional steroid administration | Adjusted steroid administration | *P* value |
|---|---|---|---|---|
| SIOH case IOP ≥ 23 mmHg, n (%) | 111(20.56) | 53 (26.37) | 58 (17.11) | 0.010 |
| SIOH case IOP ≥ 30 mmHg, n (%) | 65(12.03) | 34(16.92) | 31 (9.14) | 0.007 |
| The day postoperative that SIOH occured, mean±SD | 4.83 ± 1.89 | 5.26 ± 1.90 | 4.43 ± 1.80 | 0.020 |

SIOH = steroid-induced ocular hypertension. IOP = intraocular pressure. SD = standard deviation.

damage to the optic nerve and other tissues [6,7]. The introduction of new more accurate surgical techniques may reduce the need for post-operative steroid treatment. Therefore, in the study, we aimed to identify the risk factors for developing SIOH following vitrectomy to optimize the post-operative steroid dose for these patients.

In our study, 20.56% of patients experienced SIOH and nearly 27.9% of the patients with SIOH experienced an IOP of 40 mmHg. Some of the SIOH patients suffered from significant eye pain accompanied by headache, vomiting, and other symptoms. Moreover, some patients developed asymptomatic elevated IOP, which was detected during routine follow-up visits. Although their IOP returned to normal later, glaucomatous optic disc atrophy was found months later in the clinic work. Cheng et al. [8] reported an SIOH incidence rate of 14.6% following a 20G vitrectomy, while Brennan et al. reported a 22% elevated IOP incidence rate within two weeks after a 23G vitrectomy [9]. The slight difference in the SIOH incidence rate compared to that of our study might be due to different surgical techniques and definitions used to define high IOP. However, consistent with previous work, our findings highlight the importance of monitoring the IOP of patients treated with steroids to minimize the risk of damaging the eye [6,7].

The onset and duration of raised IOP varied between studies. Roberti et al. noted that the onset of raised IOP can occur within a few hours to several years after chronic steroid use [2]. The IOP usually returned to normal within 1–4 weeks after stopping the medication [2]. Cheng et al. [8] reported that SIOH occurred on average within 12 days post-surgery (range 4–20 days). In this study, cases of SIOH occurred 2–10 days after surgery, with about 83.78% of the cases occurring between the third and seventh day after surgery. In our study, we did not observe an onset of SIOH after 10 days post-surgery since the steroid treatment was discontinued 2 weeks after surgery. However, compared to previous studies, we observed a more rapid increase in the IOP within 24 hours. The differential response to steroids in our study may be attributable to genetic variations that influence the patients' sensitivity to steroids [10], or potentially to other, as yet unidentified, surgery-related factors.

Consistent with previous work, in our study, myopia was identified as a potential independent risk factor for SIOH [2,3,10]. Steroids may inhibit the trabecular meshwork cells' ability to phagocytose cellular debris, leading to debris blockage and subsequent obstruction of aqueous humor outflow and ultimately raised IOP [2,10]. Cho et al. found that individuals with lower trabecular meshwork height (especially <646.75 μm) are more prone to SIOH [11]. Myopic patients tend to have a lower trabecular meshwork height and a larger Schlemm's canal [12]. Therefore these patients are more at risk of developing IOP.

Silicone oil tamponade is one of the major causes of elevated IOP after vitrectomy [13]. Although the mechanism behind the development of raised IOP in patients treated with silicone oil remains unclear, several confounding factors can increase the risk of developing SIOH. The silicone oil tamponade can cause inflammation and pupillary block. In addition, the silicone oil may leak into the anterior chamber and increase the IOP. In the late postoperative stages, silicone oil emulsification [14], anterior chamber angle closure [15], and intraocular inflammation may further increase the risk of developing SIOH [16]. Previous studies have confirmed that both standard and heavy silicone oil can lead to intraocular inflammation [17]. Histopathological analysis of retinal samples indicates that silicone oil may cause a delayed type 4

hypersensitivity reaction [17]. Patients treated with silicone oil tamponade are more likely to have severe retinal detachment that requires a longer surgical duration and recovery time. Moreover, following the surgical procedure, the patient has to lie in the prone position to ensure the proper positioning of the silicone oil, enhance healing, and reduce the risk of complications. However, the prone position may increase the risk of developing IOP [18]. In the adjusted steroid regimen, despite increased systemic steroid use, the shortened duration of topical steroids was associated with a significantly lower incidence of SIOH. It is well established that both systemic and topical corticosteroids can elevate intraocular pressure [19]. However, the risk of IOP elevation is generally lower with systemic administration than with topical ocular use [20]. Moreover, systemic steroid-induced ocular hypertension tends to occur after prolonged use, with onset typically ranging from weeks to months [21], whereas topical steroid-induced IOP elevation often develops within days to a few weeks. Given the relatively short observation period in our study, the contribution of systemic steroids to IOP elevation may have been limited. Therefore, the observed reduction in SIOH incidence in the adjusted group likely reflects the dominant effect of shortened topical steroid exposure, outweighing any potential risk from concurrent systemic steroids. Nevertheless, due to the retrospective design and the concurrent use of both routes, the independent effect of systemic steroids on IOP could not be fully quantified. The inter-individual variability in steroid response observed in our study may be influenced by genetic predisposition, ocular structural characteristics, or other population-specific factors. For example, the high prevalence of myopia in East Asian populations, a key risk factor identified in our study, may partly account for the observed risk profile. Additionally, genetic variants associated with steroid sensitivity differ across populations; the N363S polymorphism, which is linked to heightened glucocorticoid sensitivity in Europeans, is extremely rare in East Asian population [22]. Further studies incorporating East Asian-specific genetic and anatomical analyses are warranted to elucidate the mechanisms underlying steroid-induced ocular hypertension in this population.

This study has some limitations that have to be acknowledged. The diagnosis of myopia was not determined by the axial length of the eye, which may have resulted in the omission of a small number of patients with mild to moderate myopia, making the estimated odds ratios for myopia conservative. The 23G vitrectomy, as a suture-free incision technique, may have a higher risk of early scleral incision leakage compared to the 20G suture-tightened incision technique [23], which may affect the accuracy of early postoperative IOP measurements. However, some studies have found no statistical difference in IOP between sutured and non-sutured 23G surgeries [9]. In addition, because some follow-up data were collected via telephone and relied on patients' recall rather than objective measurements, the possibility of recall bias and missed detection of asymptomatic IOP elevation may have been underestimated. Furthermore, given the retrospective nature of this study, causality cannot be definitively established, and the observed IOP elevations in silicone oil-filled eyes may represent a combination of steroid and silicone oil effects. Finally, this study defined postoperative raised IOP as exceeding 23 mmHg, which may have missed some patients with low baseline IOP. It should also be noted that the definition of SIOH varies in the literature; our use of ≥23 mmHg may affect comparability with studies using a higher cutoff (e.g., ≥25 mmHg). Our study presents preliminary observations suggesting that shortening the duration of corticosteroid therapy based on postoperative inflammation levels significantly reduces the occurrence of elevated intraocular pressure. Further randomized controlled trials are needed to confirm this clinical finding.

## Conclusion

Myopia and silicone oil filling were identified as potential independent risk factors for SIOH after vitrectomy. In addition, reducing the duration of topical steroid therapy was associated with a lower incidence of SIOH, particularly in high-risk patients. Further prospective studies are warranted to validate these findings.

## Acknowledgments

We would like to thank the ophthalmology medical staff involved in patient management, including surgeons, nurses, and others. We acknowledge TopEdit LLC for the linguistic editing and proofreading of this manuscript.

## Author contributions

**Conceptualization:** Xixi Yan.

**Data curation:** Hongxia Yang, Ting Chen, Zhenyu Ji.

**Project administration:** Shuanghong Jiang, Xixi Yan.

**Writing – original draft:** Shuanghong Jiang.

**Writing – review & editing:** Hongxia Yang, Ting Chen, Zhenyu Ji, Xixi Yan.

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
