## [Decision Letter · Decision Letter 0]

5 Feb 2026

Dear Dr. Yan,

Thank you for submitting your manuscript to PLOS ONE. After careful consideration, we feel that it has merit but does not fully meet PLOS ONE’s publication criteria as it currently stands. Therefore, we invite you to submit a revised version of the manuscript that addresses the points raised during the review process.

We look forward to receiving your revised manuscript.

Kind regards,

Simanta Khadka, M.D.

Academic Editor

PLOS One

Journal Requirements:

“Hubei Provincial International Science and Technology Cooperation Program（2024EHA051)”

Reviewers' comments:

Reviewer's Responses to Questions

**Comments to the Author**

1. Is the manuscript technically sound, and do the data support the conclusions?

Reviewer #1: Yes

Reviewer #2: Yes

2. Has the statistical analysis been performed appropriately and rigorously?

Reviewer #1: Yes

Reviewer #2: Yes

3. Have the authors made all data underlying the findings in their manuscript fully available?

Reviewer #1: Yes

Reviewer #2: Yes

4. Is the manuscript presented in an intelligible fashion and written in standard English?

Reviewer #1: Yes

Reviewer #2: Yes

Reviewer #1: - The exclusion criteria are appropriate overall, but clarification is needed on whether other groups (e.g., patients with family history of glaucoma or connective tissue disorders) were excluded or considered.

- Since silicone oil tamponade was strongly associated with IOP elevation, it would be helpful to add more details about how steroid-related IOP increases were differentiated from silicone oil–related mechanisms or if they have differntiate between them.

- The adjusted steroid regimen is an important and clinically meaningful intervention. However, the concurrent use of systemic steroids introduces potential confounding, since systemic and topical steroids may affect IOP differently. A more detailed discussion of this distinction would strengthen the manuscript.

- Because some follow-up was conducted by telephone and dependable on patients' memory rather than accurate data or in person visit, the possibility of recall bias and missed cases of IOP elevation cannot be excluded. Highlighting this limitation in the discussion would improve transparency

- The point about genetic predisposition to steroid response is important but feels underdeveloped. The authors could add a short note that differences in genetics, eye structure, or other population-specific factors may help explain why people respond differently to steroids, especially in an asian cohort.

- The definition of SIOH as IOP ≥23 mmHg is reasonable, but readers would benefit from a brief acknowledgment that other studies have used ≥25 mmHg, which could affect cross-study comparability.

Reviewer #2: This is a clinically relevant and well-written retrospective study addressing steroid-induced ocular hypertension (SIOH) following 23-gauge pars plana vitrectomy (PPV). The topic is important for vitreoretinal surgeons, as postoperative steroid use is routine and SIOH can be vision-threatening if unrecognized. The sample size is reasonable, the incidence data are clearly presented, and the identification of myopia and silicone oil tamponade as independent risk factors is consistent with clinical experience.

The manuscript would benefit from methodological clarification, refinement of definitions, and deeper discussion distinguishing steroid response from other post-vitrectomy IOP mechanisms, especially in silicone oil–filled eyes.

Strengths of the study:

1. Large, single-center cohort (n=540):

Provides adequate power to assess incidence and risk factors.

2. Clear clinical question:

Focused specifically on steroid-induced ocular hypertension after 23G vitrectomy, rather than postoperative IOP rise in general.

3. Practical applicability:

The comparison between traditional vs adjusted steroid regimens is highly relevant to daily vitreoretinal practice.

4. Early postoperative IOP profiling:

Identification that >80% of SIOH occurs between postoperative days 3–7 is clinically valuable.

5. Risk stratification:

Identification of myopia and silicone oil tamponade as independent risk factors aligns with existing glaucoma and vitreoretinal literature.

6. Clear outcome:

All SIOH cases normalized within one month, reassuring clinicians about reversibility with early intervention

Major Concerns & Suggestions:

1. Definition of SIOH vs Post-Vitrectomy IOP Elevation

Major conceptual issue:

• The study defines SIOH as IOP ≥23 mmHg occurring before steroid discontinuation and resolving after stopping steroids.

• However, early postoperative IOP elevation in vitrectomized eyes—especially with silicone oil—may result from:

• Overfill

• Inflammation

• Pupillary block

• Silicone oil migration

• Gas expansion (even filtered air)

Suggestion:

Explicitly acknowledge that causality cannot be definitively established in a retrospective design, particularly in silicone oil cases. Consider reframing as “steroid-associated ocular hypertension” rather than purely steroid-induced.

2. Silicone Oil as a Confounder

• Silicone oil is a strong independent cause of elevated IOP, sometimes unrelated to steroids.

• The extremely high ORs (up to 12.86) suggest silicone oil may dominate the risk model.

Suggestions:

Clarify whether:

• Oil overfill was assessed

• Anterior chamber oil migration was documented

• Consider a subgroup analysis excluding silicone oil eyes to isolate “pure” steroid responders.

3. Myopia Definition

• Myopia was not defined using axial length, which is more relevant than refractive error in vitrectomy patients.

Suggestion:

Acknowledge more clearly that axial myopia may be underrepresented, and discuss how this could bias results toward higher-risk eyes.

4. Systemic Steroid Use

• The adjusted group had longer systemic steroid exposure, which paradoxically could increase IOP risk.

Suggestion:

Discuss why systemic steroids did not increase SIOH risk, or clarify whether doses/duration were insufficient to affect IOP.

Discussion:

The discussion is strong and appropriately integrates glaucoma and vitreoretinal literature. The proposed mechanisms linking myopia and trabecular meshwork anatomy to steroid responsiveness are convincing. The section on silicone oil-related inflammation and positioning effects is particularly valuable for vitreoretinal surgeons.

Conclusion:

The conclusions are generally supported by the data, particularly regarding:

• Higher SIOH risk in myopic eyes

• Increased risk with silicone oil tamponade

• Benefit of reducing topical steroid duration in high-risk patients

However, conclusions should reflect association rather than causation.

References: correction needed

References 10 and 11 are duplicates (Kersey & Broadway).

Thank you.

.

Reviewer #1: No

Reviewer #2: **Yes:** Babu Dhanendra Chaurasiya, MDBabu Dhanendra Chaurasiya, MDBabu Dhanendra Chaurasiya, MDBabu Dhanendra Chaurasiya, MD

---

## [Author Response · Author response to Decision Letter 1]

12 Mar 2026

Dear Editor and Reviewers,

We thank the reviewers for their valuable comments and suggestions. We have carefully revised the manuscript accordingly. All changes are marked in the “Revised Manuscript with Track Changes” file. Point-by-point responses to each comment are provided in the attached “Response to Reviewers” file.

We hope the revised manuscript is now acceptable for publication in PLOS ONE.

Sincerely,

Xixi Yan

Response to Reviewers

PONE-D-25-42138

The clinical characteristics and risk factors of steroid-induced ocular hypertension following pars plana vitrectomy

PLOS One

Dear Dr. Yan,

Thank you for submitting your manuscript to PLOS ONE. After careful consideration, we feel that it has merit but does not fully meet PLOS ONE’s publication criteria as it currently stands. Therefore, we invite you to submit a revised version of the manuscript that addresses the points raised during the review process.

• A letter that responds to each point raised by the academic editor and reviewer(s). You should upload this letter as a separate file labeled 'Response to Reviewers'.

We look forward to receiving your revised manuscript.

Response to Editor: This study is a retrospective study. For detailed research methods, please refer to the "Methods" section.

Kind regards,

Simanta Khadka, M.D.

Academic Editor

PLOS One

Journal Requirements:

“Hubei Provincial International Science and Technology Cooperation Program（2024EHA051)”

We have canceled the original financial disclosure, namely "Hubei Provincial International Science and Technology Cooperation Program (2024EHA051)". One of the authors of this study, Chen Ting(T,C), received the following funding: Fund program: International Science and Technology Cooperation Program of Hubei Province (2024EHA051). Her responsibilities, as stated at the end of the manuscript, include data collection and analysis, and reading and approving the final manuscript. The funder had no role in study design, data collection and analysis, decision to publish, or preparation of the manuscript.

Thank you for the reminder. We have deposited the minimal anonymized data set necessary to replicate our study findings in a public repository. The data are available at the following DOI: 10.6084/m9.figshare.31442131. We have updated the Data Availability statement accordingly.

Thank you for your reminder. We have carefully reviewed the reviewer's comments, and there is no specific recommendation to cite any previously published works.

Thank you for your reminder. We have carefully reviewed our reference list and confirmed that no retracted papers have been cited.

Reviewers' comments:

Reviewer's Responses to Questions

Comments to the Author

1. Is the manuscript technically sound, and do the data support the conclusions?

Reviewer #1: Yes

Reviewer #2: Yes

2. Has the statistical analysis been performed appropriately and rigorously?

Reviewer #1: Yes

Reviewer #2: Yes

3. Have the authors made all data underlying the findings in their manuscript fully available?

Reviewer #1: Yes

Reviewer #2: Yes

4. Is the manuscript presented in an intelligible fashion and written in standard English?

Reviewer #1: Yes

Reviewer #2: Yes

5. Review Comments to the Author

Reviewer #1: - The exclusion criteria are appropriate overall, but clarification is needed on whether other groups (e.g., patients with family history of glaucoma or connective tissue disorders) were excluded or considered.

We thank the reviewer for the valuable comment. We acknowledge that a family history of glaucoma and a history of connective tissue diseases may potentially influence the results, which mentioned in the instruction section. Upon reviewing the medical histories of the enrolled patients, we confirmed that none reported a family history of glaucoma or connective tissue diseases. Accordingly, we have added this criterion to the patient exclusion criteria in the Methods section.

- Since silicone oil tamponade was strongly associated with IOP elevation, it would be helpful to add more details about how steroid-related IOP increases were differentiated from silicone oil–related mechanisms or if they have differntiate between them.

We thank the reviewer for their insightful comment and understanding. We fully acknowledge that silicone oil-related factors such as silicone oil entering the anterior chamber, overfilling, or emulsification are important and common causes of postoperative intraocular pressure elevation. Accordingly, such patients were excluded from our study during the design phase; however, this was not explicitly stated in the exclusion criteria. The fact that no silicone oil related interventions were required for the management of ocular hypertension in the enrolled patients further supports that these cases were not included. In response to the reviewer's comment, we have now added this criterion to the exclusion criteria in the Methods section.

- The adjusted steroid regimen is an important and clinically meaningful intervention. However, the concurrent use of systemic steroids introduces potential confounding, since systemic and topical steroids may affect IOP differently. A more detailed discussion of this distinction would strengthen the manuscript.

We thank the reviewer for this insightful comment. We fully agree that the concurrent use of systemic and topical corticosteroids in the adjusted regimen introduces a potential confounding factor, and that a more detailed discussion of their differential effects on intraocular pressure would strengthen the manuscript. In response, we have expanded the Discussion section to address this issue:

“In the adjusted steroid regimen, despite increased systemic steroid use, the shortened duration of topical steroids was associated with a significantly lower incidence of SIOH. It is well established that both systemic and topical corticosteroids can elevate intraocular pressure. However, the risk of IOP elevation is generally lower with systemic administration than with topical ocular use. Moreover, systemic steroid-induced ocular hypertension tends to occur after prolonged use, with onset typically ranging from weeks to months, whereas topical steroid-induced IOP elevation often develops within days to a few weeks. Given the relatively short observation period in our study, the contribution of systemic steroids to IOP elevation may have been limited. Therefore, the observed reduction in SIOH incidence in the adjusted group likely reflects the dominant effect of shortened topical steroid exposure, outweighing any potential risk from concurrent systemic steroids. Nevertheless, due to the retrospective design and the concurrent use of both routes, the independent effect of systemic steroids on IOP could not be fully quantified.”

- Because some follow-up was conducted by telephone and dependable on patients' memory rather than accurate data or in person visit, the possibility of recall bias and missed cases of IOP elevation cannot be excluded. Highlighting this limitation in the discussion would improve transparency

Response: We thank the reviewer for this important observation. We agree that collecting follow-up data via telephone may introduce recall bias and lead to missed detection of asymptomatic intraocular pressure elevation. In response, we have now acknowledged this limitation in the Discussion section as follows:

“In addition, because some follow-up data were collected via telephone and relied on patients' recall rather than objective measurements, some cases of asymptomatic IOP elevation may have been underestimated.”

- The point about genetic predisposition to steroid response is important but feels underdeveloped. The authors could add a short note that differences in genetics, eye structure, or other population-specific factors may help explain why people respond differently to steroids, especially in an Asian cohort.

Response: We thank the reviewer for this insightful suggestion. We agree that a brief discussion of population-specific factors would strengthen the manuscript. In response, the following sentences have been added to the Discussion section, directly after the paragraph on systemic and topical corticosteroid use:

“The inter-individual variability in steroid response observed in our study may be influenced by genetic predisposition, ocular structural characteristics, or other population-specific factors. For example, the high prevalence of myopia in East Asian populations, a key risk factor identified in our study, may partly account for the observed risk profile. Additionally, genetic variants associated with steroid sensitivity differ across populations; the N363S polymorphism, linked to heightened glucocorticoid sensitivity in Europeans, is extremely rare in East Asians. Further studies incorporating East Asian-specific genetic and anatomical analyses are warranted to elucidate the mechanisms underlying steroid-induced ocular hypertension in this population.”

- The definition of SIOH as IOP ≥23 mmHg is reasonable, but readers would benefit from a brief acknowledgment that other studies have used ≥25 mmHg, which could affect cross-study comparability.

Response: We thank the reviewer for this constructive suggestion. We agree that acknowledging the variability in the definition o

---

## [Editor Report · Decision Letter 1]

25 Mar 2026

Dear Dr.  Yan,

Thank you for submitting your manuscript to PLOS ONE. After careful consideration, we feel that it has merit but does not fully meet PLOS ONE’s publication criteria as it currently stands. Therefore, we invite you to submit a revised version of the manuscript that addresses the points raised during the review process.

As the corresponding author, your ORCID iD is verified in the submission system and will appear in the published article. PLOS supports the use of ORCID, and we encourage all coauthors to register for an ORCID iD and use it as well. Please encourage your coauthors to verify their ORCID iD within the submission system before final acceptance, as unverified ORCID iDs will not appear in the published article. *Only* the individual author can complete the verification step; PLOS staff the individual author can complete the verification step; PLOS staff the individual author can complete the verification step; PLOS staff the individual author can complete the verification step; PLOS staff *cannot* verify ORCID iDs on behalf of authors.verify ORCID iDs on behalf of authors.verify ORCID iDs on behalf of authors.verify ORCID iDs on behalf of authors.

We look forward to receiving your revised manuscript.

Kind regards,

Simanta Khadka, M.D.

Academic Editor

PLOS One

Journal Requirements:

Additional Editor Comments:

It would have been better if the author would have done point to point reply of the reviewers rather than in a paragraph. Kind request for the authors to adhere to the suggested format.

---

## [Author Response · Author response to Decision Letter 2]

27 Mar 2026

Reviewer #1:

Comment 1: The exclusion criteria are appropriate overall, but clarification is needed on whether other groups (e.g., patients with family history of glaucoma or connective tissue disorders) were excluded or considered.

Response 1: We thank the reviewer for the valuable comment. We acknowledge that a family history of glaucoma and a history of connective tissue diseases may potentially influence the results, which mentioned in the instruction section. Upon reviewing the medical histories of the enrolled patients, we confirmed that none reported a family history of glaucoma or connective tissue diseases. Accordingly, we have added this criterion to the patient exclusion criteria in the Methods section.

Comment 2: Since silicone oil tamponade was strongly associated with IOP elevation, it would be helpful to add more details about how steroid-related IOP increases were differentiated from silicone oil–related mechanisms or if they have differntiate between them.

Response 2: We thank the reviewer for their insightful comment and understanding. We fully acknowledge that silicone oil-related factors such as silicone oil entering the anterior chamber, overfilling, or emulsification are important and common causes of postoperative intraocular pressure elevation. Accordingly, such patients were excluded from our study during the design phase; however, this was not explicitly stated in the exclusion criteria. The fact that no silicone oil related interventions were required for the management of ocular hypertension in the enrolled patients further supports that these cases were not included. In response to the reviewer's comment, we have now added this criterion to the exclusion criteria in the Methods section.

Comment 3: The adjusted steroid regimen is an important and clinically meaningful intervention. However, the concurrent use of systemic steroids introduces potential confounding, since systemic and topical steroids may affect IOP differently. A more detailed discussion of this distinction would strengthen the manuscript.

Response 3: We thank the reviewer for this insightful comment. We fully agree that the concurrent use of systemic and topical corticosteroids in the adjusted regimen introduces a potential confounding factor, and that a more detailed discussion of their differential effects on intraocular pressure would strengthen the manuscript. In response, we have expanded the Discussion section to address this issue:

“In the adjusted steroid regimen, despite increased systemic steroid use, the shortened duration of topical steroids was associated with a significantly lower incidence of SIOH. It is well established that both systemic and topical corticosteroids can elevate intraocular pressure. However, the risk of IOP elevation is generally lower with systemic administration than with topical ocular use. Moreover, systemic steroid-induced ocular hypertension tends to occur after prolonged use, with onset typically ranging from weeks to months, whereas topical steroid-induced IOP elevation often develops within days to a few weeks. Given the relatively short observation period in our study, the contribution of systemic steroids to IOP elevation may have been limited. Therefore, the observed reduction in SIOH incidence in the adjusted group likely reflects the dominant effect of shortened topical steroid exposure, outweighing any potential risk from concurrent systemic steroids. Nevertheless, due to the retrospective design and the concurrent use of both routes, the independent effect of systemic steroids on IOP could not be fully quantified.”

Comment 4: Because some follow-up was conducted by telephone and dependable on patients' memory rather than accurate data or in person visit, the possibility of recall bias and missed cases of IOP elevation cannot be excluded. Highlighting this limitation in the discussion would improve transparency

Response 4: We thank the reviewer for this important observation. We agree that collecting follow-up data via telephone may introduce recall bias and lead to missed detection of asymptomatic intraocular pressure elevation. In response, we have now acknowledged this limitation in the Discussion section as follows:

“In addition, because some follow-up data were collected via telephone and relied on patients' recall rather than objective measurements, some cases of asymptomatic IOP elevation may have been underestimated.”

Comment 5: The point about genetic predisposition to steroid response is important but feels underdeveloped. The authors could add a short note that differences in genetics, eye structure, or other population-specific factors may help explain why people respond differently to steroids, especially in an Asian cohort.

Response 5: We thank the reviewer for this insightful suggestion. We agree that a brief discussion of population-specific factors would strengthen the manuscript. In response, the following sentences have been added to the Discussion section, directly after the paragraph on systemic and topical corticosteroid use:

“The inter-individual variability in steroid response observed in our study may be influenced by genetic predisposition, ocular structural characteristics, or other population-specific factors. For example, the high prevalence of myopia in East Asian populations, a key risk factor identified in our study, may partly account for the observed risk profile. Additionally, genetic variants associated with steroid sensitivity differ across populations; the N363S polymorphism, linked to heightened glucocorticoid sensitivity in Europeans, is extremely rare in East Asians. Further studies incorporating East Asian-specific genetic and anatomical analyses are warranted to elucidate the mechanisms underlying steroid-induced ocular hypertension in this population.”

Comment 6: The definition of SIOH as IOP ≥23 mmHg is reasonable, but readers would benefit from a brief acknowledgment that other studies have used ≥25 mmHg, which could affect cross-study comparability.

Response 6: We thank the reviewer for this constructive suggestion. We agree that acknowledging the variability in the definition of SIOH across studies would enhance the transparency and interpretability of our findings. In response, we have added the following sentence to the Limitations section:

“It should also be noted that the definition of SIOH varies in the literature; our use of ≥23 mmHg may affect comparability with studies using a higher cutoff (e.g., ≥25 mmHg).”

We believe this addition appropriately addresses the reviewer’s concern and helps readers better contextualize our findings.

Reviewer #2: This is a clinically relevant and well-written retrospective study addressing steroid-induced ocular hypertension (SIOH) following 23-gauge pars plana vitrectomy (PPV). The topic is important for vitreoretinal surgeons, as postoperative steroid use is routine and SIOH can be vision-threatening if unrecognized. The sample size is reasonable, the incidence data are clearly presented, and the identification of myopia and silicone oil tamponade as independent risk factors is consistent with clinical experience.

The manuscript would benefit from methodological clarification, refinement of definitions, and deeper discussion distinguishing steroid response from other post-vitrectomy IOP mechanisms, especially in silicone oil–filled eyes.

Strengths of the study:

1. Large, single-center cohort (n=540):

Provides adequate power to assess incidence and risk factors.

2. Clear clinical question:

Focused specifically on steroid-induced ocular hypertension after 23G vitrectomy, rather than postoperative IOP rise in general.

3. Practical applicability:

The comparison between traditional vs adjusted steroid regimens is highly relevant to daily vitreoretinal practice.

4. Early postoperative IOP profiling:

Identification that >80% of SIOH occurs between postoperative days 3–7 is clinically valuable.

5. Risk stratification:

Identification of myopia and silicone oil tamponade as independent risk factors aligns with existing glaucoma and vitreoretinal literature.

6. Clear outcome:

All SIOH cases normalized within one month, reassuring clinicians about reversibility with early intervention

Comment 1:

Major Concerns & Suggestions:

1. Definition of SIOH vs Post-Vitrectomy IOP Elevation

Major conceptual issue:

• The study defines SIOH as IOP ≥23 mmHg occurring before steroid discontinuation and resolving after stopping steroids.

• However, early postoperative IOP elevation in vitrectomized eyes—especially with silicone oil—may result from:

• Overfill

• Inflammation

• Pupillary block

• Silicone oil migration

• Gas expansion (even filtered air)

Suggestion:

Explicitly acknowledge that causality cannot be definitively established in a retrospective design, particularly in silicone oil cases. Consider reframing as “steroid-associated ocular hypertension” rather than purely steroid-induced.

Response 1: We thank the reviewer for raising this important conceptual issue. We fully agree that early postoperative IOP elevation in vitrectomized eyes can result from multiple factors unrelated to steroid use, and that distinguishing steroid-induced ocular hypertension from these other causes is critical to the validity of our findings. We also acknowledge that causality cannot be definitively established in a retrospective design, and that the term “steroid-associated” may be more precise from a strictly epidemiological perspective. However, given the widespread use of “steroid-induced” in the literature and the temporal relationship observed in our study, we have retained the term SIOH while explicitly acknowledging this limitation in the revised manuscript. To address this concern, we have taken the following measures:

First, we have clarified in the revised Methods section that patients with silicone oil‑related factors (e.g., silicone oil entering the anterior chamber, overfilling, or emulsification) and gas expansion occurring during the predefined follow-up period were excluded from the study. It is important to note that these patients were already excluded during the initial data collection phase based on clinical judgment; the revision merely makes this criterion explicit in the manuscript. Therefore, the study population and all analyses remain unchanged. (A similar clarification was also provided in response to a related comment from Reviewer #1.)

Second, other known causes of postoperative IOP elevation—such as exudative anterior uveitis, pupillary block, and anterior chamber or vitreous hemorrhage—were already specified in the exclusion criteria and were excluded accordingly.

Third, the diagnosis of SIOH in our study required documented IOP elevation during steroid use and resolution after withdrawal. While not definitive proof, this temporal relationship provides supportive evidence for a steroid-induced component.

We acknowledge that in a retrospective study, complete separation of these factors is challenging. However, by applying strict exclusion criteria and considering the temporal relationship with steroid use, we have endeavored to minimize confounding, and the cases identified as SIOH in our study are likely to represent true steroid-induced ocular hypertension in the majority of instances.

Comment 2: Silicone Oil as a Confounder

• Silicone oil is a strong independent cause of elevated IOP, sometimes unrelated to steroids.

• The extremely high ORs (up to 12.86) suggest silicone oil may dominate the risk model.

Suggestions:

Clarify whether:

• Oil overfill was assessed

• Anterior chamber oil migration was documented

• Consider a subgroup analysis excluding silicone oil eyes to isolate “pure” steroid responders.

Response 2: We thank the reviewer for this important comment. We fully acknowledge that silicone oil tamponade is a potent independent risk factor for postoperative IOP elevation, and that its strong effect (OR up to 12.86 in our study) could potentially confound the assessment of steroid-induced ocular hypertension. We have addressed this concern as follows:

1. Assessment of silicone oil overfill and anterior chamber migration

In our clinical practice, silicone oil overfill and anterior chamber migration are routinely documented during postoperative slit-lamp examinations. Cases with clear evidence of these mechanical complications during the predefined follow-up period were excluded from the study. In the revised Methods section, we have now explicitly added "silicone oil‑related factors (e.g., silicone oil entering the anterior chamber, overfilling, or emulsification)" to the exclusion criteria to make this transparent.

As discussed in our manuscript, silicone oil tamponade can contribute to IOP elevation through multiple mechanisms, including mechanical factors (e.g., overfill, anterior chamber migration), inflammation, emulsification, angle closure, and even the required prone positioning. Additionally, patients receiving silicone oil often have more severe underlying retinal pathology, which may independently influence postoperative IOP dynamics.

To address whether silicone oil dominated the risk model, we performed a subgroup analysis in patients without silicone oil tamponade who received the traditional steroid regimen. Due to the reduced sample size after excluding silicone oil eyes (n=193 overall; n=66 in the traditional steroid subgroup), the number of events available for multivariate analysis was limited. For the outcome of IOP ≥23 mmHg, 55 patients were included in the multivariate model, and myopia remained a strong risk factor with an OR of 14.8 (95%CI:0.58–375.08, p=0.10). For the outcome of IOP ≥30 mmHg, only 18 patients were available for analysis, which precluded meaningful multivariate regression. This further underscores the challenge of isolating steroid-specific effects in a retrospective cohort where silicone oil use is prevalent. These findings highlight the need for future studies with larger cohorts of non-silicone oil‑filled vitrectomy patients to further validate steroid-specific risk factors.

Comment 3: Myopia Definition

• Myopia was not defined using axial length, which is more relevant than refractive error in vitrectomy patients.

Suggestion:

Acknowledge more clearly that axial myopia may be underrepresented, and discuss how this could bias results toward higher-risk eyes.

Response 3: We thank the reviewer for this insightful comment. We fully agree that axial length is a more accurate measure of myopia than refractive error. This could lead to an underestimation of the association between myopia and SIOH, as some high-risk eyes were inadvertently included in the reference group. We have now clarified this point in the Limitations section as follows:

“The diagnosis of myopia was not determined by the axial length of the eye, which may have resulted in the omission of a small number of patients with mild to moderate myopia, making the estimated odds ratios for myopia conservative.”

Comment 4: Systemic Steroid Use

• The adjusted group had longer systemic steroid exposure, which paradoxically could increase IOP risk.

Suggestion:

Discuss why systemic steroids did not increase SIOH risk, or clarify whether doses/duration were insufficient to affect IOP.

Response 4: we thank the reviewer for this important observation. In response, we have expanded the Discussion section to address why increased systemic steroid exposure in the adjusted group did not lead to a higher risk of SIOH. As stated in the revised manuscript:

“In the adjusted steroid regimen, despite increased systemic steroid use, the shortened duration of topical steroids was associated with a significantly lower incidence of SIOH. It is well established that both systemic and topical corticosteroids can elevate intraocular pressure [19]. However, the risk of IOP elevation is generally lower with systemic administration than with topical ocular use [20]. Moreover, systemic steroid-induced ocular hypertension tends to occur after prolonged use, with onset typically ranging from weeks to months [21], whereas topical steroid-induced IOP elevation often develops within days to a few weeks. Given the relatively short observation period in our study, the c

---

## [Editor Report · Decision Letter 2]

5 Apr 2026

The clinical characteristics and risk factors of steroid-induced ocular hypertension following pars plana vitrectomy

PONE-D-25-42138R2

Dear Dr. Xixi,

We’re pleased to inform you that your manuscript has been judged scientifically suitable for publication and will be formally accepted for publication once it meets all outstanding technical requirements.

Kind regards,

Simanta Khadka, M.D.

Academic Editor

PLOS One
---

## [Editor Report · Acceptance letter]

PONE-D-25-42138R2

PLOS One

Dear Dr. Yan,

I'm pleased to inform you that your manuscript has been deemed suitable for publication in PLOS One. Congratulations! Your manuscript is now being handed over to our production team.

Kind regards,

on behalf of

Dr. Simanta Khadka

Academic Editor

PLOS One